# Sterol regulatory element binding protein 1 couples mechanical cues and lipid metabolism

Rebecca Bertolio [1,2], Francesco Napoletano[1,2], Miguel Mano [3], Sebastian Maurer-Stroh [4,5], Marco Fantuz[1,6], Alessandro Zannini[1,2], Silvio Bicciato [7], Giovanni Sorrentino[1,9] & Giannino Del Sal [1,2,8]

Sterol regulatory element binding proteins (SREBPs) are a family of transcription factors that regulate lipid biosynthesis and adipogenesis by controlling the expression of several enzymes required for cholesterol, fatty acid, triacylglycerol and phospholipid synthesis. In vertebrates, SREBP activation is mainly controlled by a complex and well-characterized feedback mechanism mediated by cholesterol, a crucial bio-product of the SREBP-activated mevalonate pathway. In this work, we identified acto-myosin contractility and mechanical forces imposed by the extracellular matrix (ECM) as SREBP1 regulators. SREBP1 control by mechanical cues depends on geranylgeranyl pyrophosphate, another key bio-product of the mevalonate pathway, and impacts on stem cell fate in mouse and on fat storage in *Drosophila*. Mechanistically, we show that activation of AMP-activated protein kinase (AMPK) by ECM stiffening and geranylgeranylated RhoA-dependent acto-myosin contraction inhibits SREBP1 activation. Our results unveil an unpredicted and evolutionary conserved role of SREBP1 in rewiring cell metabolism in response to mechanical cues.

[1] Laboratorio Nazionale CIB, Area Science Park, Padriciano 99, Trieste, Italy. [2] Dipartimento di Scienze della Vita, Università degli Studi di Trieste, Trieste, Italy. [3] Center for Neuroscience and Cell Biology, University of Coimbra, Coimbra, Portugal. [4] Bioinformatics Institute (BII), Agency for Science Technology and Research (A*STAR), 30 Biopolis Street, #07-01 Matrix, Singapore 138671, Singapore. [5] Department of Biological Sciences (DBS), National University of Singapore (NUS), 14 Science Drive 4, Singapore 117543, Singapore. [6] International School for Advanced Studies (SISSA), Trieste, Italy. [7] Department of Life Sciences, University of Modena and Reggio Emilia, Modena, Italy. [8] IFOM, the FIRC Institute of Molecular Oncology, Via Adamello, 16-20139 Milan, Italy. [9] Present address: Laboratory of Metabolic Signaling, Institute of Bioengineering, Ecole Polytechnique Fédérale de Lausanne, CH-1015 Lausanne, Switzerland. Correspondence and requests for materials should be addressed to G.S. (email: giovanni.sorrentino@epfl.ch) or to G.D.S. (email: gdelsal@units.it)

Conserved from yeast to humans, SREBPs are transcription factors which function as master regulators of genes that control cellular lipid homoeostasis to meet organismal requirements[1]. SREBPs have been shown to play key roles in coupling lipid metabolism with nutrition, cell growth, energy stress, inflammation and other physiological and pathological processes[2]. In vertebrates, SREBP1 and SREBP2 proteins translocate from the endoplasmic reticulum to the Golgi apparatus, in response to deprivation of cholesterol. Then they are processed by proteolytic cleavage and targeted to the nucleus, where they induce the expression of genes mainly involved in sterol and fatty acid biosynthesis[3].

The complex function of SREBP proteins implicates a fine-tuning of their activity at multiple levels. In addition to mechanisms that control the stability of SREBP proteins through a complex repertoire of post-translational modifications (PTMs), negative feedback loops by the end products of the SREBP pathway play a key role in regulating SREBP activation[2]. While SREBP translocation and proteolytic maturation mechanisms are widely conserved, the auto-regulatory systems show differences among species[4]. For example, SREBP activity is primarily sensitive to phospholipids in *Drosophila melanogaster* and *Caenorhabditis elegans*, which are cholesterol auxotrophs. Conversely, SREBP activity is primarily sensitive to sterols in vertebrates, though SREBP1 maturation has been shown to be sensitive to some polyunsaturated fatty acids (PUFAs) in human cells, independently of cholesterol[5].

In addition to sterols, the SREBP-induced mevalonate pathway produces key isoprenoid metabolites, farnesyl pyrophosphate (FPP) and geranylgeranyl pyrophosphate (GGPP), which serve as lipid donors for regulatory PTMs of target proteins (prenylation)[6]. Inhibition of SREBPs, by reducing the levels of isoprenoids, impacts on a plethora of biological processes, including cell division, migration, death, intracellular trafficking, protein stability and cytoskeleton organization[7]. Deregulation of isoprenoid levels and protein prenylation is involved in many pathological conditions such as neurodegeneration, cardiovascular diseases and cancer[6,8–10]. Despite their crucial activity as pathophysiological effectors of SREBPs, the role of isoprenoids in the regulation of SREBP activation has been poorly investigated so far.

Here we demonstrate that acto-myosin contraction, downstream of RhoA prenylation, controls SREBP1 function, a mechanism conserved from *Drosophila* to human that links extracellular matrix (ECM) mechanical cues to lipid metabolism. Mechanistically we find that, in response to ECM rigidity, the energy sensor AMP-activated protein kinase (AMPK) inhibits SREBP1 activation, which impacts on physiological and pathophysiological processes such as mesenchymal stem cell (MSC) differentiation and tissue fibrosis.

## Results

**Protein geranylgeranylation controls SREBP1 transcriptional activity**. To investigate whether isoprenoids play a role in the activation of SREBPs, human epithelial breast cell lines were transfected with two reporter plasmids, low density lipoprotein promoter-luciferase (LDLR-Luc)[11] and Steaoryl-CoA desaturase promoter-luciferase (SCD1-Luc), as readouts of SREBP activation and were maintained in conditions of reduced intracellular cholesterol in order to activate SREBPs. Specifically, cells were treated with cerivastatin, or grown in serum-free or lipid-depleted media. All these conditions induced a robust activation of SREBPs, as demonstrated by increased luciferase activity after 24 h, using either LDLR-Luc (Fig. 1a) or SCD1-Luc (Supplementary Fig. 1a). As expected, supplementing the medium with cholesterol prevented SREBP activation (Fig. 1a and Supplementary Fig. 1a). Interestingly, addition of GGPP to the medium, but not of FPP,

inhibited SREBP activation to an extent comparable to cholesterol addition (Fig. 1a and Supplementary Fig. 1a). These results were confirmed by analysing the expression in serum-starved cells of four endogenous SREBP target genes, *LDLR*, *SCD1*, *Acetyl-CoA Carboxylase 1 (ACC1)* and *Fatty Acid Synthase (FASN)* at the mRNA levels (Fig. 1b), and of SCD1 protein level (Fig. 1c). The processing of SREBP1 was strongly prevented by GGPP in serum-starved cells after 24 h of treatment, while under the same conditions SREBP2 processing remained unaltered (Fig. 1c). To completely deprive cells of cholesterol, both exogenously uptaken and endogenously synthetized, cells were maintained in lipid-depleted medium and treated with statin. In these conditions, GGPP addition prevented activation of LDLR-Luc (Fig. 1d) and SCD1-Luc (Supplementary Fig. 1b), upregulation of *LDLR*, *SCD1*, *ACC1* and *FASN* mRNA (Supplementary Fig. 1c), of SCD1 protein (Supplementary Fig. 1d), and processing of SREBP1 (Supplementary Fig. 1d). This result clearly demonstrates that the effect of GGPP was independent of cholesterol.

The effect of GGPP on SREBP1 processing was further confirmed in a panel of cell lines from different tissues, as breast (MDA-MB 231), lung (H1299) and liver (Mahlavu and Immortalized Human Hepatocytes, namely IHH) (Supplementary Fig. 1e, f). In line with these results, GGPP treatment prevented SREBP1 nuclear translocation induced by serum starvation (Supplementary Fig. 1g). Altogether, these results establish GGPP, a key intermediate of the mevalonate pathway, as an endogenous modulator of SREBP1 maturation and function.

Upon binding to geranylgeranyl-transferase 1 (GGTI, hereafter referred to as GGTase1), GGPP modifies a considerable number of target proteins (Fig. 1e) mainly involved in signal transduction, structural organization and trafficking, controlling their localization and function[6]. To test whether protein geranylgeranylation was involved in SREBP1 activation, we inhibited the transfer of the geranylgeranyl moiety to target proteins by using GGTI-298, a specific inhibitor of GGTase1[9]. This treatment induced a strong activation of SREBP1 transcriptional activity, as assessed using the LDLR-Luc (Fig. 1f and Supplementary Fig. 1h) and SCD1-Luc (Supplementary Fig. 1i) reporters. This effect was specific since a mutation of a single nucleotide within the sterol responsive element (SRE) of the LDLR-Luc reporter construct completely prevented the luciferase signal (Fig. 1f). GGTI-298 treatment also caused a robust and time-dependent increase of SREBP1 maturation (Fig. 1g and Supplementary Fig. 1k), with a consequent induction of SREBP1 target genes, as monitored by upregulation of *SCD1* mRNA (Fig. 1h and Supplementary Fig. 2l) and protein (Fig. 1g and Supplementary Fig. 1k) levels, and upregulation of *LDLR*, *ACC1*, *FASN*, and *3-hydroxy-3-methylglutaryl-CoA reductase (HMGCR)* mRNA levels (Fig. 1h, Supplementary Fig. 1j and Supplementary Fig. 2l). This effect was prevented by SREBP1 knockdown (Fig. 1i and Supplementary Fig. 1l). SREBP1 activation upon GGTI-298 treatment also increased the intracellular content of lipids, with a consequent accumulation of lipid droplets within the cytoplasm (Fig. 1j). Under these conditions, adding back GGPP did not reverse SREBP1 activation (Supplementary Fig. 1m), suggesting that protein prenylation, instead of the GGPP intracellular levels, is required for SREBP1 regulation.

Taken together, these data demonstrate that SREBP1 is controlled by protein geranylgeranylation and suggest that one or more GGTase1 target proteins are involved in SREBP1 activation and lipid biosynthesis.

**Evolutionarily conserved SREBP inhibition by RhoA and acto-myosin contraction**. In humans, a total of 124 proteins (including isoforms), corresponding to 75 genes[12,13], are predicted to be

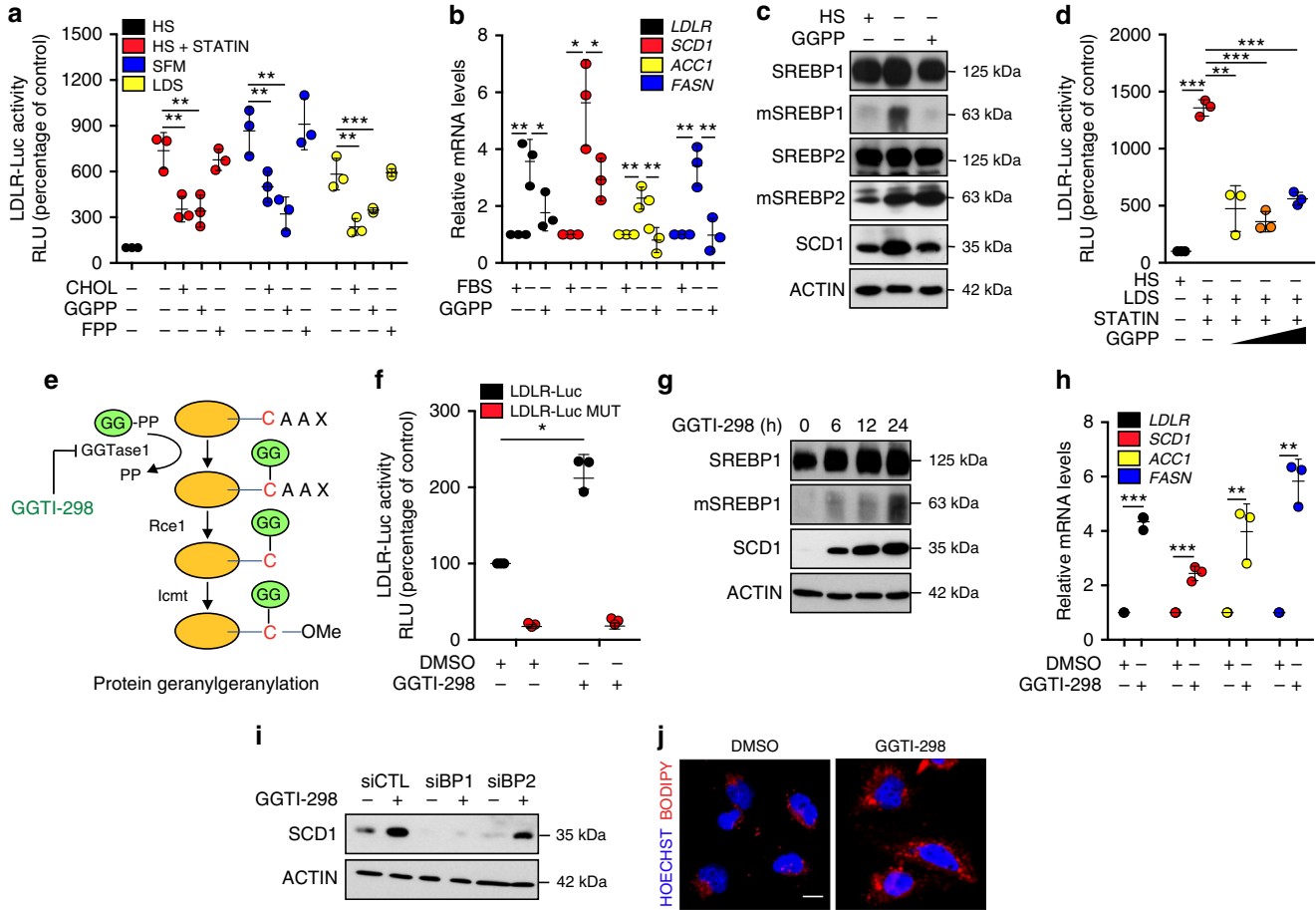

**Fig. 1** Protein geranylgeranylation regulates SREBP1. **a** Low density lipoprotein receptor promoter-luciferase (LDLR-Luc) assay in MCF-10A cells. Medium containing 5% horse serum (HS, as control) was replaced with 5% HS medium supplemented with 10 μM cerivastatin (STATIN), serum-free medium (SFM) or 2% lipid serum (lipid-depleted serum, LDS) medium, for 24 h. Cells were either mock-treated, or treated with cholesterol (CHOL), geranylgeranyl pyrophosphate (GGPP) or farnesyl pyrophosphate (FPP). **b** RT-qPCR quantification of *LDLR*, *SCD1*, *ACC1* and *FASN* gene expression in MCF-10A cells. **c** Western blot analysis of MCF-10A cells. **d** LDLR-Luc assay in MCF-10A cells. 5% HS medium (control) was replaced with medium supplemented with 2% LDS and 1 μM cerivastatin (STATIN), and increasing doses of GGPP (20, 40 and 100 μM) for 24 h. **e** Scheme of geranylgeranyl (GG) conjugation to cysteine. **f** LDLR-Luc assay in MCF-10A cells treated with DMSO as control or geranylgeranyl pyrophosphate transferase I inhibitor (GGTI-298). Cells transfected with the mutated construct LDLR-Luc MUT underwent the same treatments. **g** Western blot analysis of MCF-10A cells treated with GGTI-298 for the indicated time (hours, h). **h** RT-qPCR quantification of *LDLR*, *SCD1, ACC1 and FASN* gene expression in MCF-10A cells treated with DMSO as control or GGTI-298. **i** Western blot analysis of MCF-10A cells transfected with control (siCTL) SREBP1 (siBP1) and SREBP2 (siBP2) siRNAs and treated with GGTI-298 for 24 h. **j** BODIPY 493/503 staining of lipid droplets (in red) in Mahlavu cells treated with GGTI-298. Nuclei were stained with HOECHST (in blue). Scale bar, 15 μm. Graph bars represent mean ± s.d. of *n* = 3 biological replicates. Values in (**a**, **b** and **f**) are expressed as Relative Luminometer Units (RLU). Values in (**b** and **h**) are expressed as mRNA levels relative to control. In (**b**, **c**) 5% HS medium (control) was replaced with SFM or SFM supplemented with GGPP for 24 h. For western blots, ACTIN was used as loading control, mSREBP indicates mature protein. Blots and images are representative of *n* = 3 biological replicates. *P* value: *$P < 0.05$, **$P < 0.01$, ***$P < 0.001$ by two-tailed Student's *t*-test for all analyses

geranylgeranylated by GGTase1. To identify geranylgeranylated proteins involved in SREBP1 regulation, we assembled a custom library of siRNAs targeting all but one (PALM3) of the mRNAs coding for proteins predicted to be GGTase1 targets. Next, we screened this library in MDA-MB-231 cells, in condition of basal SREBP activity (medium supplemented with 10% Fetal Bovine Serum, FBS), using a LDLR-Luc-based SREBP1 activation assay. As shown in Fig. 2a, the majority of transfected siRNAs led to SREBP activation. Of note, many of the targeted proteins, whose down-regulation by siRNA was able to activate SREBP1, were involved in actin cytoskeleton dynamics (Supplementary Fig. 2a). Among these proteins, RhoA, a small GTPase known to control acto-myosin dynamics, scored as second best hit in the screening (Fig. 2a and Supplementary Table 3) and the first one after validation of the top five hits using independent siRNAs and LDLR-Luc or SCD1-Luc reporter (Supplementary Fig. 2b–d)[14]. This evidence led us to

hypothesise that RhoA could act as a negative regulator of SREBP1 and lipid biosynthesis. We next evaluated the level of RhoA pre-nylation in cells in which statin treatment inhibited mevalonate pathway, leading to SREBP1 activation. In these conditions, a reduction of RhoA prenylation correlated with the activation of SREBP1 (Fig. 2c), while addition of GGPP to the medium efficiently rescued RhoA prenylation (Fig. 2c). This suggested that RhoA activation might play a role on SREBP1 regulation. Confirming this hypothesis, silencing of RhoA by two independent siRNAs (Supplementary Fig. 2b) caused SREBP1 maturation and functional activation, as assessed by SCD1 protein accumulation (Fig. 2b) and either LDLR-Luc or SCD1-Luc reporter activity (Supplementary Fig. 2c, d). Conversely, overexpression of a constitutively active form of RhoA, RhoA G14V (Supplementary Fig. 2g), caused a reduction of LDLR-Luc (Fig. 2d and Supplementary Fig. 2h) signal and SCD1-Luc (Supplementary Fig. 2i), while overexpression of the

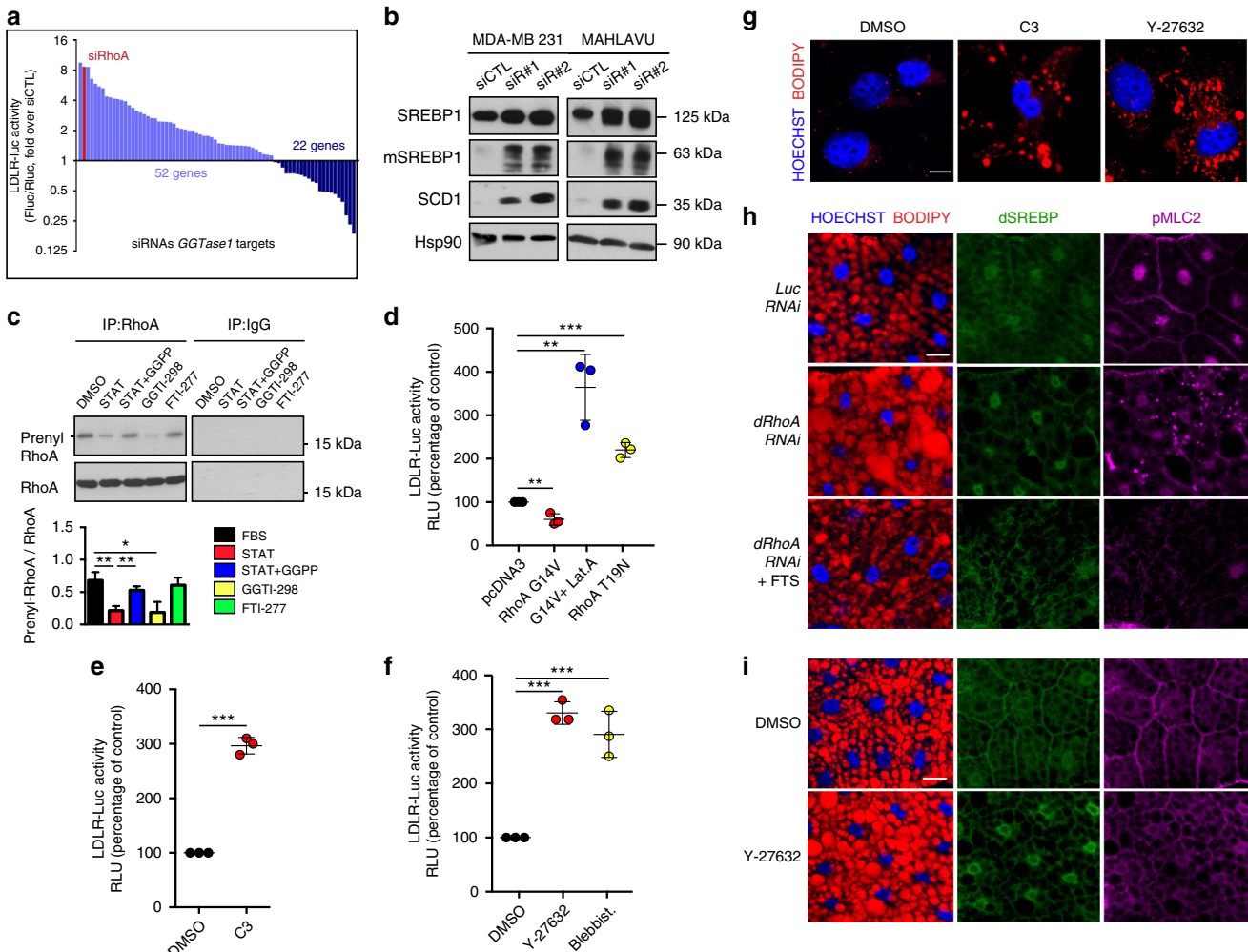

**Fig. 2** RhoA and acto-myosin regulate the activity of hSREBP1 and dSREBP. **a** Screening of low density lipoprotein receptor promoter-luciferase activity in MDA-MB-231 cells transfected with constructs expressing firefly (LDLR-Luc) and renilla (Rluc) luciferase and either control siRNA (siCTL) or siRNAs targeting genes encoding geranylgeranyl pyrophosphate transferase I (GGTase1) protein substrates. **b** Western blot analysis of MDA-MB-231 and Mahlavu cells 48 h after transfection with siCTL or *RhoA* targeting siRNAs (siR#1 and siR#2). Hsp90 was used as loading control. mSREBP indicates mature protein. **c** Western blot analyses of immunoprecipitated (IP) RhoA in MCF-10A cells treated DMSO (as control), 1 μM cerivastatin (STAT), 1 μM cerivastatin and 20 μM GGPP (STAT+GGPP), 5 μM GGTI-298 or 5 μM FTI-277. IgGs were used as IP control. **d** LDLR-Luc assay in MCF-10A cells 12 h after transfection with pcDNA3-GFP control plasmid, pcDNA3-GFP-RhoA G14V construct with a 6 h DMSO treatment, pcDNA3-GFP-RhoA G14V construct with a 6 h Latrunculin A (G14V + Lat.A) treatment, or pcDNA3-GFP-RhoA T19N construct. **e** LDLR-Luc assay in MCF-10A cells treated with either DMSO as control or C3 for 24 h. **f** LDLR-Luc assay in MCF-10A cells treated with either DMSO as control, Y-27632, or Blebbistatin (Blebbist.) for 24 h. **g** BODIPY 493/503 staining of lipid droplets (in red) in Mahlavu cells treated with either DMSO, C3 or Y-27632 for 24 h. Scale bar, 15 μm. **h, i** BODIPY 493/503 staining of lipid droplets (in red) and immunofluorescence analysis of dSREBP (in green) and phosphorylated myosin light chain (pMLC2, in magenta) in *Drosophila* larval fat body from **h** flies expressing either Luciferase or *dRhoA RNAi*, or *dRhoA RNAi* and treated with fatostatin (FTS) and **i** wild-type flies treated with either DMSO or Y-27632. Scale bar, 20 μm. Graphs bars in (**c–f**) represent mean ± s.d. of n = 3 biological replicates. Values in (**d–f**) are expressed as Relative Luminometer Units (RLU). Nuclei in (**h–i**) were stained with HOECHST (in blue). Blots and images are representative of n = 3 biological replicates. *P* value: *P < 0.05, **P < 0.01, ***P < 0.001 by two-tailed Student's *t*-test for all analyses

dominant negative RhoA T19N (Supplementary Fig. 2g) caused LDLR-Luc (Fig. 2d and Supplementary Fig. 2h) and SCD1-Luc activation (Supplementary Fig. 2i). Accordingly, RhoA inhibition by C3 toxin caused LDLR-Luc (Fig. 2e and Supplementary Fig. 2j) and SCD1-Luc activation (Supplementary Fig. 2k) and induction of *LDLR*, *SCD1* and *HMGCR* mRNA expression (Supplementary Fig. 2l).

Upon geranylgeranylation, RhoA localizes at the plasma membrane and triggers actin polymerisation and acto-myosin contraction, via activation of its downstream effector Rho-kinase (ROCK)[15], which phosphorylates the myosin light chain 2 (MLC2). To test whether the regulation of SREBP1 by RhoA requires the canonical ROCK pathway, we treated human

epithelial breast cells with either the ROCK inhibitor Y-27632 or the myosin inhibitor Blebbistatin and monitored SREBP1 transcriptional activity. Both treatments activated SREBP1, as shown by induction of either LDLR-Luc (Fig. 2f and Supplementary Fig. 2j) or SCD1-Luc reporter activity (Supplementary Fig. 2k), and by *LDLR*, *SCD1*, *ACC1*, *FASN* and *HMGCR* mRNA expression (Supplementary Fig. 2l). Moreover, the effect of RhoA overexpression on SREBP1 activation was abolished by inhibition of actin polymerisation with Latrunculin A treatment (Fig. 2d and Supplementary Fig. 2h, i), thus demonstrating that actin polymerisation acts downstream of RhoA/ROCK signalling to control SREBP1 activation. Furthermore, inhibition of either RhoA or ROCK induced a massive accumulation of lipid droplets

in liver cells (Fig. 2g). Taken together, these experiments show that RhoA/ROCK signalling and acto-myosin dynamics impact on SREBP1 activation and lipid biosynthesis in human cells.

We further investigated the impact of the ROCK/acto-myosin axis on SREBP activity, by using *Drosophila melanogaster* as a model organism, taking advantage that it expresses a single SREBP homologue (dSREBP)[4] whose activity is controlled independently of cholesterol[16,17]. dSREBP turned out to be activated in *Drosophila* S2 cells maintained for 24 h in serum-free medium. This effect was prevented by supplementing the medium with GGPP (Supplementary Fig. 3a). Moreover, inhibition of either actin polymerisation or acto-myosin contraction, with Latrunculin A or the ROCK inhibitor Y-27632, respectively, caused activation of dSREBP to an extent similar to maintaining cells in lipid-depleted medium[16] (Supplementary Fig. 3b), thus demonstrating that the ROCK/acto-myosin axis controls SREBP maturation in *Drosophila* cells. We next validated these findings in vivo, by monitoring dSREBP activation and lipid accumulation in the larval adipose tissue (fat body)[18], upon either tissue-specific knockdown of *Drosophila* RhoA (dRhoA), or inhibition of ROCK by Y-27632[19,20]. dRhoA knockdown and ROCK inhibition impaired acto-myosin contractility in adipocytes, as demonstrated by reduced staining of phosphorylated MLC2 (pMLC2) (Fig. 2h, i). Strikingly, both conditions promoted dSREBP maturation (Supplementary Fig. 3c, d), nuclear translocation (Fig. 2h, i) and transcriptional activity in these cells, as shown by upregulation of mRNA levels of the dSREBP target *Fatty Acid Synthase* (FAS) (Supplementary Fig. 3e) and expansion of LDs (Fig. 2h, i and Supplementary Fig. 3f, g). Treatment with fatostatin, a potent SREBP1 inhibitor[21], suppressed both phenotypes induced by dRhoA knockdown, i.e. FAS upregulation (Supplementary Fig. 3e) and lipid accumulation (Fig. 2h), demonstrating that these effects were the result of dSREBP activity.

Altogether, these experiments demonstrate that modulation of acto-myosin dynamics impacts on dSREBP activation and lipid biosynthesis and accumulation in a living organism.

**ECM stiffening inhibits SREBP-dependent lipid biosynthesis.** Acto-myosin contractility has a key role in intracellular sensing and transduction of mechanical forces generated by the architecture and rigidity of the ECM. The physical properties of the ECM influence the growth and shape of virtually all tissues and organs[22], and impact on a plethora of processes ranging from tissue morphogenesis to cancer development. Physical cues are promptly sensed by acto-myosin through RhoA[22]. Therefore, we hypothesized that the physical features of the ECM could impact on SREBP1 activation via RhoA. We tested this hypothesis by growing human epithelial breast cells on fibronectin-coated hydrogels characterized by progressively reduced elastic moduli (e.g. 50, 4 and 0.5 kPa; Fig. 3a). In these conditions, ECM softening led to a marked impairment of mechano-signalling pathways, as demonstrated by reduction of phosphorylated MLC2 (pMLC2) and Focal Adhesion Kinase (pFAK), as well as of the protein levels of the mechano-transducer TAZ (Fig. 3b, e), and triggered a progressive induction of SREBP1 protein maturation (Fig. 3b, e) and transcriptional activity (Fig. 3c, d and Supplementary Fig. 4a, b). Furthermore, cells grown in soft ECM showed a marked SREBP1-dependent lipid droplets accumulation (Fig. 3d, f and Supplementary Fig. 4b). In line with these data, unbiased gene set enrichment analysis showed that the genes involved in lipid metabolism were enriched in MDA-MB-231 cells grown on soft as compared to cells grown on stiff hydrogels (Supplementary Fig. 4c)[23]. Using the same culturing conditions, the impact of mechanical stimuli on SREBP1 cleavage and activity

was further confirmed in primary cells from mouse mammary and liver epithelia, and in cell lines from different human tissue origin (i.e. liver, colon, breast, pancreas and others). Indeed, along with reduced pMLC2 and TAZ protein levels, all the cells grown on soft conditions exhibited an increased maturation of SREBP1 and, an enhancement of its transcriptional activity, as demonstrated by increased SCD1 protein levels (Supplementary Fig. 4d).

Taken together, these results demonstrate that SREBP1 activity is under control of mechanical cues and that increase of ECM stiffness may inhibit its biological function.

To validate our observations in human samples, we generated a signature specific for SREBP1[24] (Supplementary Table 4), as a proxy of SREBP1 activation, to interrogate two public datasets of human transcriptional profiles from different physiological and pathological conditions: (i) normal breast tissue with high vs low mammographic density[25], a parameter that directly correlates with tissue stiffness[26]; (ii) lung tissue from patients affected by Idiopathic Pulmonary Fibrosis, a condition linked to pulmonary tissue stiffness[27], vs healthy controls[28]. Of note, activation of SREBP1 inversely correlated with mammographic density and lung tissue fibrosis, suggesting that in these physio-pathological conditions SREBP1 might respond to mechanical cues (Fig. 3g).

**AMPK suppresses SREBP1 activation downstream of mechanical inputs.** We next investigated how mechanical stimuli and acto-myosin dynamics control SREBP1 activation. SREBP1 maturation, stability, nuclear accumulation and transcriptional activity are controlled by a complex repertoire of PTMs[29–32]. Among them, phosphorylation by AMP-activated protein kinase (AMPK), a key sensor of the cellular energy status, inhibits SREBP1 proteolytic maturation[29]. AMPK has been recently found to be activated by E-cadherin-dependent actin polymerisation[33] and by mechanosensitive $Ca^{2+}$ influx at focal adhesions[34]. Therefore, we reasoned that, in response to an increase in ECM stiffness, acto-myosin contraction leads to AMPK activation and consequent SREBP1 inhibition, as a mechanism to prevent anabolic processes and increase ATP availability. Consistently, in human breast epithelial cells, inhibition of RhoA/ROCK by either siRNA, treatment with different drugs (C3 and Y-27632), or soft ECM culture conditions, suppressed AMPK activation, as demonstrated by reduced phosphorylation of its activation loop (pThr172) and its target protein Acetyl-CoA Carboxylase (ACC1, Fig. 4a–c). Importantly, in human epithelial breast cells either treated with Y-27632 or grown on soft ECM, reactivation of AMPK by AICAR, an AMP analogue known to activate AMPK, inhibited SREBP1 maturation, SCD1 transcriptional induction and lipid droplet accumulation (Fig. 4d–f), indicating that AMPK mediates the effects of mechanical forces on SREBP1.These data demonstrate that AMPK coordinates SREBP1 activation and lipid biosynthesis downstream of environmental mechanical cues

**SREBP1 drives mechano-dependent MSC adipogenic commitment.** Based on our evidence, it is conceivable that, through this mechanism, soft substrates could promote adipogenesis by unleashing SREBP1 activity upon inhibition of acto-myosin contraction[35,36]. This could be the case of MSCs. Indeed, adult MSCs are known to spontaneously differentiate into adipocytes or osteoblasts when cultured on a soft or stiff matrix, respectively[37,38]. However, the mechanisms by which a soft microenvironment instructs MSCs to undergo adipogenesis are largely unknown. To test our hypothesis, we cultured mMSCs on either stiff or soft fibronectin-coated hydrogels, or we treated mMSCs cultured in stiff matrix with Y-27632. In these conditions, we monitored adipogenesis by evaluating mRNA levels of the adipogenesis markers *Pparg, AdipoQ, Fabp4* and *Cebpa* (Fig. 5a, d) and by Oil-Red-O

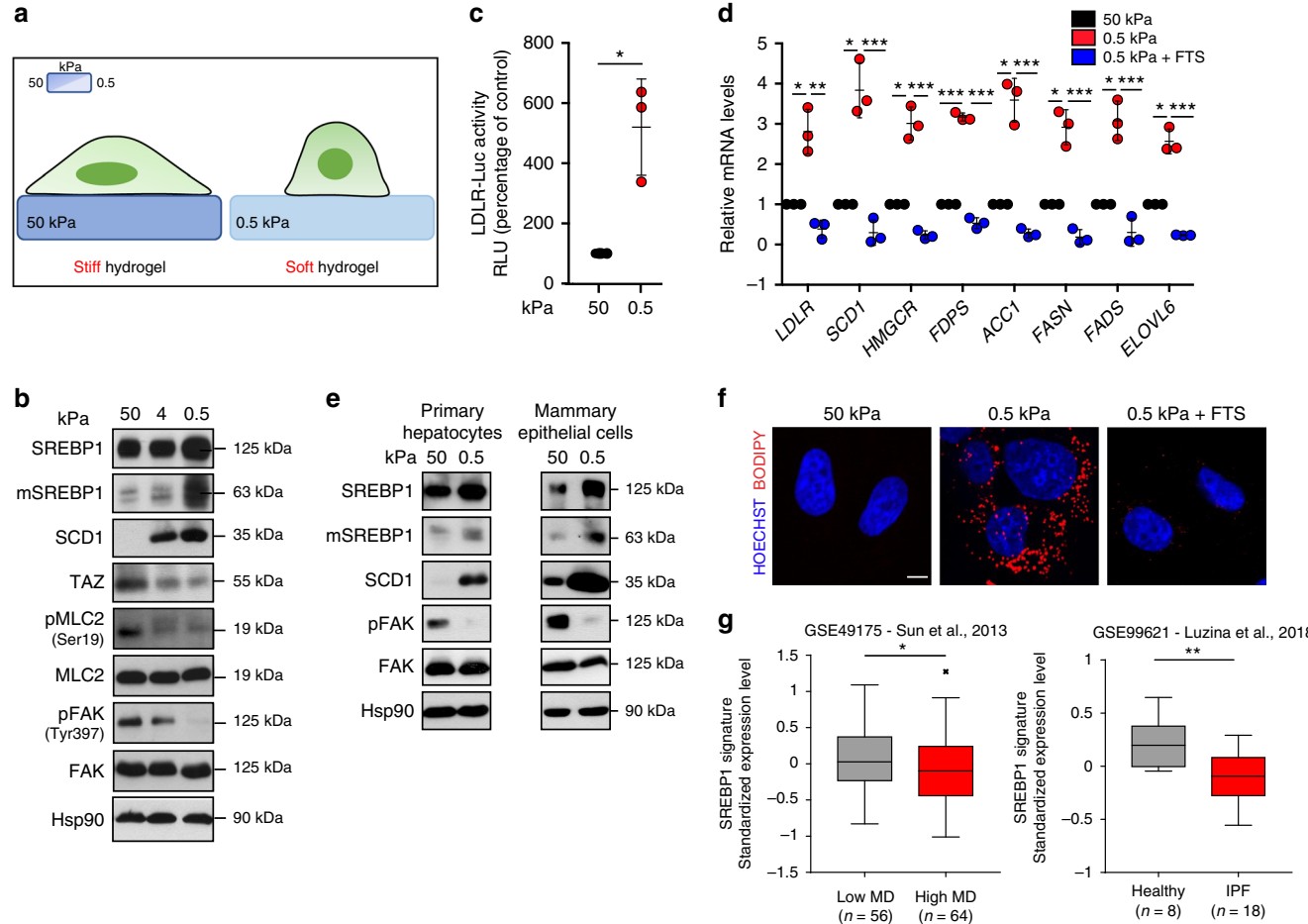

**Fig. 3** SREBP1 is under the control of mechanical cues. **a** Scheme of cell (in green) adhesion to fibronectin-coated hydrogel matrix (in blue) with 50 and 0.5 kPa elastic modulus. **b** Western blot analysis of MCF-10A cells cultured on fibronectin-coated hydrogel matrix of high (50 kPa elastic modulus), intermediate (4 kPa elastic modulus) or low (0.5 kPa elastic modulus) stiffness, for 24 h. **c** Low density lipoprotein receptor promoter-luciferase (LDLR-Luc) assay in MCF-10A cells cultured on stiff (50 kPa elastic modulus, as control) or soft (0.5 kPa elastic modulus) fibronectin-coated hydrogel matrix for 24 h. **d** RT-qPCR quantification of the expression of the indicated genes in MCF-10A cells cultured on stiff (50 kPa elastic modulus, as control) or soft (0.5 kPa elastic modulus) fibronectin-coated hydrogel matrix, or soft (0.5 kPa elastic modulus) fibronectin-coated hydrogel matrix with fatostatin treatment (FTS), for 24 h. **e** Western blot analysis of primary cultures of hepatocytes and mammary epithelial cells on stiff (50 kPa elastic modulus) or soft (0.5 kPa elastic modulus) fibronectin-coated hydrogel matrix for 24 h. **f** BODIPY 493/503 staining of lipid droplets (in red) in Mahlavu cells cultured on stiff (50 kPa elastic modulus, as control, in black) or soft (0.5 kPa elastic modulus, in red) fibronectin-coated hydrogel matrix, or soft (0.5 kPa elastic modulus) fibronectin-coated hydrogel matrix with fatostatin treatment (FTS, in blue), for 24 h. Nuclei were stained with HOECHST (in blue). Scale bar, 10 μm. **g** Average standardized expression levels of SREBP1 signature in human samples datasets of normal breast tissue with high vs low mammographic density (MD) and Idiopathic Pulmonary Fibrosis (IPF) patients vs healthy controls. Graphs bars represent mean ± s.d. of $n = 3$ biological replicates. $P$ value: $*P < 0.05$ by two-tailed Student's $t$-test (**c**, **d**) and two-way ANOVA (**g**). For western blots, Hsp90 was used as loading control, mSREBP indicates mature protein. Blots and images are representative of $n = 3$ biological replicates

staining (Fig. 5b, e). As expected, adipogenesis was significantly induced by ECM softening (Fig. 5a, b) or by ROCK inhibition in stiff substrates (Fig. 5d, e), thus confirming a key role of ECM rigidity in MSC fate determination. Importantly, in mMSCs grown on soft ECM or treated with the ROCK inhibitor, AMPK activity was reduced and SREBP1 was strongly activated (Fig. 5c, f), while SREBP1 pharmacological inhibition by fatostatin (Fig. 5c, f) prevented MSC differentiation into adipocytes (Fig. 5a, b, d, e). These results demonstrate that SREBP1 in response to mechanical cues drives adipogenic specification of MSCs.

## Discussion

In this work, we highlighted a cholesterol-independent feedback mechanism of SREBP1 regulation, mediated by geranylgeranyl pyrophosphate. This isoprenoid is a mevalonate-derived metabolite exerting crucial functions in regulating cell homoeostasis, mainly through protein geranylgeranylation. Geranylgeranylation of several Rho-GTPases is required to coordinate acto-myosin dynamics in response to chemical and physical stimuli. We demonstrated that prenylation of RhoA strongly inhibits SREBP1 activity, thus establishing a crucial role of SREBP1 as a transducer of environmental mechanical cues and unveiling a link between the physical properties of the ECM and lipid metabolism. SREBP1 inhibition exerted by RhoA turned out to be conserved from invertebrates to humans, and it may represent a fundamental mechanism to regulate cell behaviour. Supporting this notion, we showed that SREBP1 activation determines the fate of MSCs in response to mechanical cues.

Furthermore, we found that acto-myosin contractility regulates *Drosophila* fat storage via SREBP-dependent lipid biosynthesis and accumulation, supporting that coordination between mechanical and metabolic pathways may have implications at the organismal

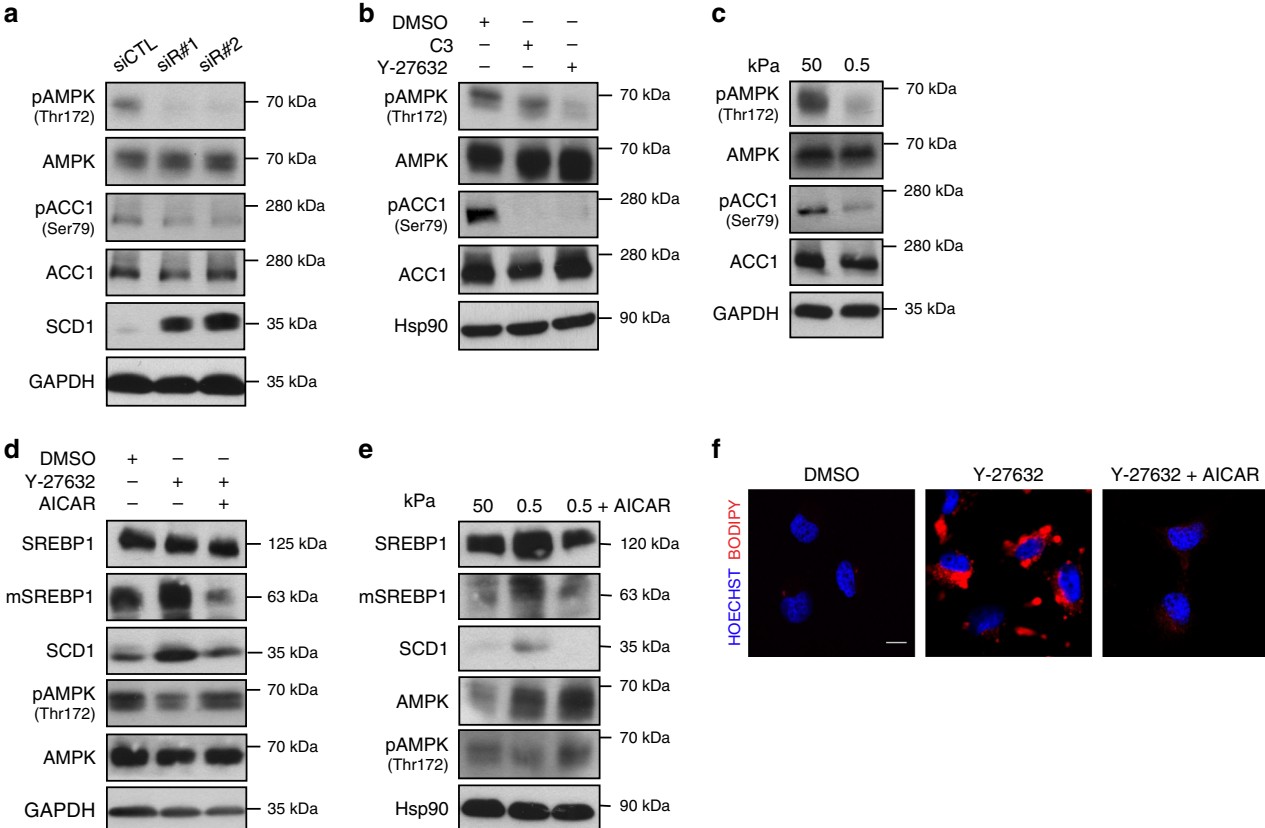

**Fig. 4** AMPK suppresses SREBP1 activation downstream of mechanical inputs. **a** Western blot analysis of MCF-10A cells 48 h after transfection with either control (siCTL) or *RhoA* targeting siRNA (siR#1 and siR#2). GAPDH was used as loading control. **b** Western blot analysis of MCF-10A cells treated with either DMSO, C3 or Y-27632, for 24 h. Hsp90 was used as loading control. **c** Western blot analysis MCF-10A cells cultured on either stiff (50 kPa elastic modulus) or soft (0.5 kPa elastic modulus) fibronectin-coated hydrogel matrix for 24 h. GAPDH was used as loading control. **d** Western blot analysis of MCF-10A cells treated with either DMSO, Y-27632 or AICAR, for 24 h. GAPDH was used as loading control. **e** Western blot analysis of MCF-10A cells cultured on either stiff (50 kPa elastic modulus) or soft (0.5 kPa elastic modulus) fibronectin-coated hydrogel matrix, or soft (0.5 kPa elastic modulus) fibronectin-coated hydrogel matrix with AICAR treatment, for 24 h. Hsp90 was used as loading control. **f** BODIPY 493/503 staining of lipid droplets (in red) in Mahlavu cells treated with Y-27632 or Y-27632 and AICAR, for 24 h. Nuclei were stained with HOECHST (in blue). Scale bar, 15 µm. For western blots, mSREBP indicates mature protein. Blots and images are representative of $n = 3$ biological replicates

level such as in human physio-pathologic conditions, which is shown by our results (Fig. 3g) in breast and lung tissues with altered ECM deposition and stiffening. In other diseases, as in cancer, this homoeostatic interplay might be overcome at multiple levels. Indeed, diverse oncogenic proteins have been shown to be activated and stabilized upon ECM stiffening and, in turn, to directly engage key metabolic transducers to boost cancer cell proliferation[23,39–41].

Acto-myosin dynamics has been estimated to drain an important fraction of cellular energy[42], and it has been recently proposed to stimulate glucose uptake and ATP production via AMPK[43], as well as to foster glycolysis via the release of aldolase A from actin fibres[44]. Our finding that SREBP1-dependent lipid anabolism is prevented by cytoskeleton contraction via activation of AMPK suggests that this mechanism could be important to integrate environmental mechanical signals with intracellular requirements, allowing a cell to maintain an optimal energy status.

## Methods

**Cell lines**. MCF-10A cells are a human immortalized normal epithelial breast cell line and were cultured in Dulbecco's Modified Eagle's Medium (DMEM)/F12 (LONZA) (1:1) supplemented with 5% Horse Serum (HS), 100 U/mL penicillin and 10 µg/mL streptomycin, 20 ng/ml recombinant human epidermal growth factor (EGF), 10 µg/ml recombinant human insulin and 500 ng/ml hydrocortisone. MDA-MB-231 are a human breast cancer cell line. IHH are immortalized human

hepatocytes. Mahlavu are human hepatocellular carcinoma cells. HT29 are human colorectal adenocarcinoma cells. U87MG and U251 are human glioblastoma cells. U2OS are a human osteosarcoma cells line. H1299 are a human non-small cell lung cancer cell line. PANC-1 are a human pancreatic adenocarcinoma cell line. U87MG and U251 are glioblastoma cell lines. HT29, MDA-MB-231, PANC-1, U2OS, U87MG and U251 cells were cultured in DMEM (LONZA) supplemented with 10% Foetal Bovine Serum (FBS) 100 U/mL penicillin and 10 µg/mL strepto-mycin. Mahlavu cells were cultured in Eagle's Minimum Essential Medium (EMEM, Sigma) supplemented with FBS, 100 U/mL penicillin, 10 µg/mL strepto-mycin, 1% Minimum Essential Medium Non-Essential Amino Acids (MEM NEAA) and 1% Glutamax. IHH cells were cultured in DMEM/F12 (LONZA) (1:1) supplemented with 10% FBS, 100 U/mL penicillin and 10 µg/mL streptomycin, 5 µg/ml recombinant human insulin, 1 µg/ml hydrocortisone and 1% Glutamax. H1299 cells were cultured in RPMI 1640 supplemented with 10% FBS, 100 U/mL penicillin and 10 µg/mL streptomycin.

Primary mouse mammary epithelial cells were cultured in DMEM supplemented with 10% FBS, 100 U/mL penicillin and 10 µg/mL streptomycin, 20 ng/ml recombinant human EGF, 10 µg/ml recombinant human insulin, 500 ng/ml hydrocortisone. Primary hepatocytes were cultured in DMEM supplemented with 10% FBS, 100 U/mL penicillin, 10 µg/mL streptomycin, 1% glutamine, 6 µg/ml insulin and 1 µM dexamethasone.

Human cell lines are from ATCC or other laboratories cooperating on the project. *Drosophila melanogaster* Schneider's 2 (S2) cell line was a kind gift from F. Feiguin, International Centre for Genetic Engineering and Biotechnology (ICGEB), Trieste.

Cells were subjected to STR genotyping with PowerPlex 18D System and confirmed in their identity comparing the results to reference cell databases (DMSZ, ATCC and JCRB databases).

Cells were tested for mycoplasma contamination with negative results.

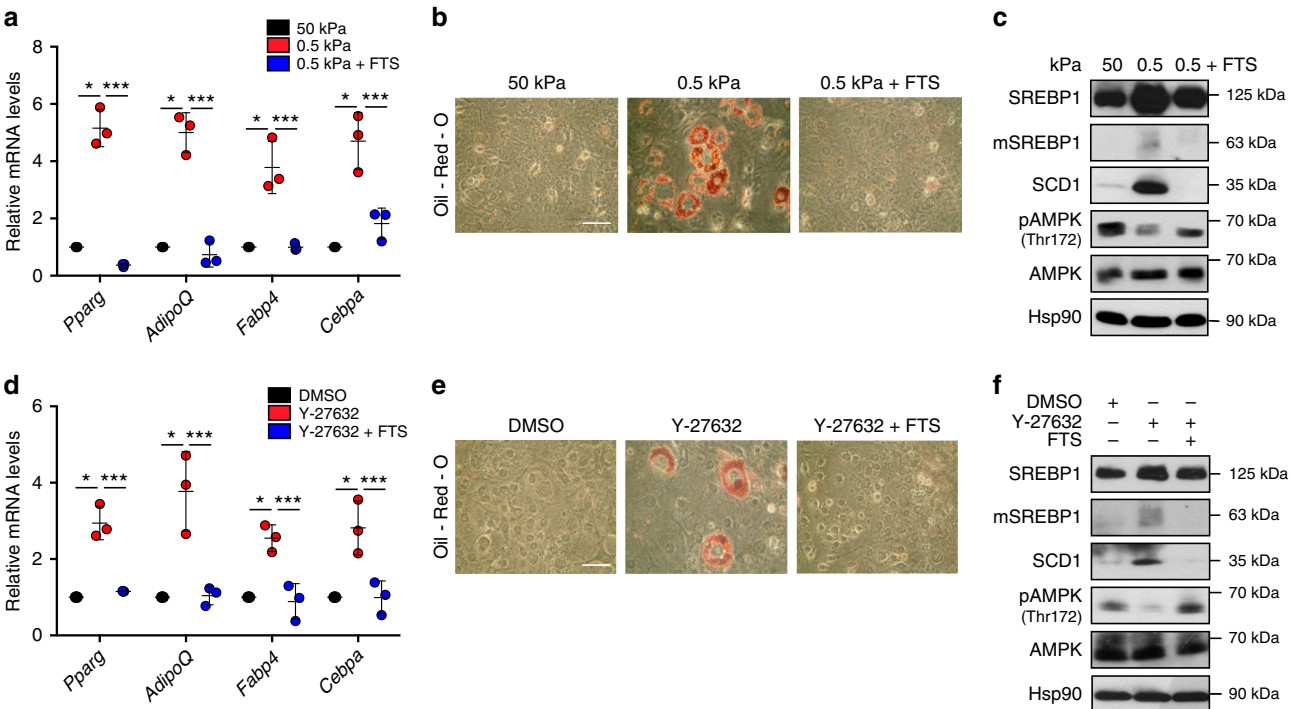

**Fig. 5** SREBP1 regulates mesenchymal stem cell commitment upon cytoskeleton remodelling. **a** RT-qPCR quantification of the indicated genes in mouse mesenchymal stem cells (mMSCs) cultured on stiff (50 kPa elastic modulus, as control) or soft (0.5 kPa elastic modulus) fibronectin-coated hydrogel matrix, or soft (0.5 kPa elastic modulus) fibronectin-coated hydrogel matrix with fatostatin treatment (FTS). Values are expressed as mRNA levels relative to control. **b** Oil-Red-O staining of lipid droplets (in red) in mMSCs cultured on either stiff (50 kPa elastic modulus) or soft (0.5 kPa elastic modulus) fibronectin-coated hydrogel matrix, or soft (0.5 kPa elastic modulus) fibronectin-coated hydrogel matrix with fatostatin treatment (FTS). Scale bar, 100 μm. **c** Western blot analysis of mMSCs cultured on either stiff (50 kPa elastic modulus) or soft (0.5 kPa elastic modulus) fibronectin-coated hydrogel matrix, or soft (0.5 kPa elastic modulus) fibronectin-coated hydrogel matrix with fatostatin treatment (FTS). **d** RT-qPCR quantification of the indicated genes in mMSCs cultured in differentiating medium and treated with either DMSO as control, or with Y-27632, or Y-27632 and fatostatin (Y-27632 + FTS). Values are expressed as mRNA levels relative to control. **e** Oil-Red-O staining of lipid droplets (in red) in mMSCs cultured in differentiating medium and treated with DMSO as control, or with Y-27632 or Y-27632 and fatostatin (Y-27632+FTS). Medium composition is detailed in the Methods section. Scale bar, 100 μm. **f** Western blot analysis of mMSC cultured in differentiating medium and treated with either DMSO as control or with Y-27632, or Y-27632 and fatostatin (FTS). Graph bars represent mean ± s.d. of $n = 3$ biological replicates. For western blots, Hsp90 was used as loading control, mSREBP indicates mature protein. P value: *$P < 0.05$, ***$P < 0.001$ by two-tailed paired Student's t-test. Blots and images are representative of $n = 3$ biological replicates

**Preparation of fibronectin-coated hydrogel matrix**. 50, 4 or 0.5 kPa Easy Coat hydrogels (Cell guidance system) were coated with 10 μg/ml fibronectin.

**Reagents and plasmids**. The following compounds and working concentration were used: AICAR (1 mM, Sigma Aldrich A9978), Blebbistatin (50 μM Sigma Aldrich B0560), Cerivastatin (1 μM Sigma Aldrich SML0005), Cholesterol (0.5 mM, Sigma Aldrich C8667), Farnesyl Pyrophosphate (20 μM, Sigma Aldrich F6892), Fatostatin hydrocloryde (20 μM, Sigma Aldrich F8932), Fibronectin (10 μg/ml, Sigma Aldrich F0895), GGTI-298 (5 μM, Sigma Aldrich G5169), Geranylgeranyl Pyrophosphate (20 μM, Sigma Aldrich G6025), Y-27632 (20 μM, Sigma Aldrich Y0503). Latrunculin A (0.5 μM, Santa Cruz Biotechnologies sc-202691) and C3 (100 ng/ml, Cytoskeleton CT04). DMSO was purchased from Sigma Aldrich (D4540). Lipoprotein Depleted Serum was purchased from Biowest (S181L). Treatments lasted 24 h unless otherwise stated.

pEGFP–RhoA-G14V was a gift from C. Schneider (Laboratorio Nazionale CIB, Italy). pcDNA3-EGFP-RhoA-T19N (Addgene plasmid #12967) was a gift from Gary Bokoch. The pLDLR-Luc construct (also known as pES7, Addgene plasmids #14940), harbouring the SREBP-responsive Sterol Responsive Element (SRE) sequence (ATCACCCCAC), and the pLDLR-Luc mutSRE construct (LDLR-Luc MUT, Addgene plasmid #14945), harbouring a SREBP-unresponsive mutant SRE (ATAACCCCAC)[11] were gifts from Axel Nohturfft. The pGL3-SCD1-Luc construct was generated by cloning a PCR amplified DNA fragment corresponding to nucleotides −405 to −229 of the human SCD1 gene into the pGL3 vector with *KpnI* and *BglII* restriction enzymes.

**Transfections**. siRNA transfections were performed with Lipofectamine RNAi-MAX (Life technologies) in antibiotic-free medium according to manufacturer instructions. Sequences of siRNAs are reported in Supplementary Table 1. Control siRNA was AllStars negative control Qiagen 1027281. Plasmid DNA transfections

of MDA-MB-231 and MCF-10A cells were performed with LTX (Invitrogen) in antibiotic-free medium according to the manufacturer instructions.

**Luciferase assay**. pLDLR-Luc (300 ng/cm²), pLDLR-Luc mutSRE (300 ng/cm²)[11], and pGL3-SCD1-Luc reporters (300 ng/cm²) were co-transfected with the CMV–Renilla construct (30 ng/cm²), 12 h after transfection with either siRNAs, pcDNA3 (100 ng/cm²), pcDNA3–GFP-RhoA-G14V (100 ng/cm²) or pcDNA3-GFP-RhoA-T19N plasmids (100 ng/cm²). Luciferase/Renilla signal was analysed in cell lysates, 24 h after transfection of luciferase reporters, using the Dual-Luciferase Reporter Assay System (Promega E1910).

**Quantitative RT-PCR**. Cells were harvested in Qiazol lyses reagent (Qiagen) for total RNA extraction, and contaminant DNA was removed by DNase treatment. Quantitative real time PCR analyses were carried out on cDNAs retrotranscribed with iScript™ Advanced cDNA Synthesis Kit (Biorad 172-5038) and analysed with BIORAD CFX96 Touch™ detection system and Biorad CFX Manager software. Quantitative analysis was performed by the 2$^{-\Delta\Delta Ct}$ method. *Histone 3* was used as reference gene. PCR primer sequences are reported in Supplementary Table 2.

**Antibodies**. The following antibodies and working concentrations were used: anti-Actin C11 (1:5000, Sigma Aldrich A2066, for western blot), anti-SREBP1 2A4 (1:500, Santa Cruz Biotechnology sc13551, for western blot), anti-SREBP1 H160 (1:100, Santa Cruz Biotechnology sc8984, for immunofluorescence), anti-SREBP2 (1:500, BD Bioscience 557037, for western blot), anti-SCD1 (1:1000, Abcam ab19862, for western blot), anti-GAPDH (6C5) (1:5000, Santa Cruz Biotechnology sc32233, for western blot), anti-AMPK (1:1000, Cell Signalling 2532S, for western blot), anti-AMPK phospho Thr172 (1:1000, Cell Signalling 2531S, for western blot), anti-ACC1 (1:1000, Cell Signalling 3676S, for western blot), anti-ACC1

phospho Ser79 (1:1000, (Cell Signalling 11818S, for western blot), anti-Farnesyl (1:1000, AB4073 Merck Millipore), anti-Hsp90 (1:2000, Santa Cruz Biotechnology sc13119, for western blot), anti-MLC2 (1:1000, Cell Signalling 3675S, for western blot), anti-MLC2 phospho Ser19 (1:500, Cell Signalling 3671S, for western blot and immunofluorescence), anti-FAK C-20 (1:1000, Santa Cruz Biotechnology sc-558, for western blot) and anti-FAK phospho Y397 (1:1000, Abcam ab81298, for western blot).

**Targeted RNAi screening.** Seventy four siRNAs targeting all mRNAs coding for human proteins predicted to be GGTI targets (Supplementary Table 3) were cherry-picked robotically from a human genome-wide siRNA library (siGENOME SMARTPool; pools of four siRNAs per gene, Dharmacon) and arrayed in 384-well white plates (PerkinElmer). For the screening experiments, MDA-MB-231 cells were transfected with the targeted siRNA library, using a standard reverse transfection protocol[45]. Briefly, the transfection reagent (Lipofectamine RNAiMAX; Life Technologies) was diluted in OPTI-MEM (Life Technologies) and added to the siRNAs arrayed on the 384-well plates; 30 min later, the cells were suspended in culture medium without antibiotics and seeded ($2.0 \times 10^3$ per well). Twenty-four hours after siRNA transfection, cells were transfected with the LDLR-Luc reporter (Firefly), together with a plasmid expressing Renilla luciferase for normalization, using FuGENE HD Transfection Reagent (Promega) diluted in OPTI-MEM; the transfection mix (10 μl per well) was added to cells using a Multidrop Combi Reagent Dispenser (Thermo Fisher Scientific). Forty-eight hours after transfection of the reporter (i.e. 72 h after siRNA transfection), the cells were lysed in 1× Glo Lysis Buffer (Promega); Firefly and Renilla luciferase activities were measured using the Dual-Glo Luciferase Assay System (Promega), according to the manufacturer's instructions, using an Envision Multimode Plate Reader (PerkinElmer). Normalized LDLR reporter activity was calculated as fold change over a control non-targeting siRNA.

**Western blot analysis of mammalian and _Drosophila_ S2 cells.** For immuno-blotting analyses protein were lysed in Lysis Buffer (NP40 1%, Tris-HCL pH = 8 50 mM, NaCl 150 mM, EDTA 1 mM) solution, supplemented with protease and phosphatase inhibitors. Lysates were loaded and separated in SDS-PAGE, followed by western blotting on Nitrocellulose membranes (Amersham). Blocking was performed in Blotto-tween (PBS, 0.2% Tween-20, not fat dry milk 5%) or with TBST (0.2% Tween-20, Tris/HCl 25 mM pH 7.5) plus 5% BSA (PanReac Applichem) depending on the antibody.

**Immunoprecipitation.** Immunoprecipitation (IP) experiments were performed using IP buffer (NaCl 120 mM, Tris-HCl pH8 20 mM, EDTA 1 mM, NP40 0,5%) with protease inhibitors. Samples were sonicated three times and cleared by centrifugation for 10 min at 13,000 × g at 4 °C and incubated for 4 h at 4 °C with anti-RhoA or IgG antibodies. After 1 h incubation with protein G-Sepharose (GE Healthcare), immunoprecipitates were washed three times in IP buffer, resuspended in sample buffer and analyzed by immunoblotting.

**Immunofluorescence analysis of mammalian cells.** Briefly, cells were fixed in 4% paraformaldehyde for 10 min, washed in phosphate buffered saline (PBS), permeabilized with 0.1% Triton X-100 for 10 min and blocked in 3% FBS/PBS for 30 min. Antigen recognition was done by incubation with primary antibody at 4 °C for 14 h and with secondary antibodies (goat anti-mouse Alexa Fluor 568 and goat anti-rabbit Alexa Fluor 488, Life Technologies) at 4 °C for 2 h. Nuclei were stained with Hoechst 33342 (Life Technologies) for 15 min.

**BODIPY staining of mammalian cells.** Cells were fixed in 4% paraformaldehyde for 15 min, stained with 0.1 μg/ml BODIPY-493/503 at 37 °C for 30 min and DAPI at 25 °C for 15 min. Cells were imaged using a Nikon Ti-E inverted fluorescence microscope.

**_Drosophila_ cell lines.** _Drosophila melanogaster_ Schneider's 2 (S2) line cells are macrophage-like cells of embryonic origin (kind gift from F. Feiguin, International Centre for Genetic Engineering and Biotechnology, Trieste) and were cultured in Insect-XPRESS medium (LONZA 12-730) supplemented with heat-inactivated 10% FBS, 100 U/mL penicillin and 10 μg/mL streptomycin, at 25 °C.

**_Drosophila_ lines and housing.** The following _Drosophila_ lines were used: $w^{1118}$ (as wild-type control, kind gift of F. Feiguin, International Centre for Genetic Engineering and Biotechnology, Trieste, Italy), $w^{1118};UAS-dRhoA^{RNAi}/CyO-GFP$ (generated from line VDRC#12734, Vienna Drosophila Resource Center), $w^{1118};cg-Gal4$ (Bloomington Drosophila Stock Center BDSC#7011, kind gift of P. Bellosta, University of Trento, Italy) and $w^{1118};UAS-Luciferase^{RNAi}$ (Bloomington Drosophila Stock Center BDSC#35788, kind gift of B. Mollereau, Ecole Normale Superierure de Lyon, France). The genotype of larvae analysed in Figs. 2h and 3c, e, f was: $w^{1118};cg-Gal4;UAS-Luciferase^{RNAi}$ (_Luc RNAi_) and $w^{1118};cg-Gal4/UAS-dRhoA^{RNAi}$ (_dRhoA RNAi_). Flies were maintained at 25 °C on standard corn/yeast medium.

**Dissection and culture of _Drosophila_ larval fat bodies.** Fat bodies were dissected from $N = 6$ third instar female larvae in PBS and either immediately processed or cultured for 14 h in Insect-XPRESS medium (LONZA 12-730) supplemented with heat-inactivated 10% FBS, 100 U/mL penicillin, 10 μg/mL streptomycin and 10 μg/ml Insulin and drug (Y-27632, Fatostatin or AICAR) or drug solvent (according to the manufacturer instructions).

**Western blot analysis of _Drosophila_ fat bodies.** Fat bodies freshly dissected in PBS and cultured fat bodies washed in PBS were transferred in lyses buffer and processed for western blot analysis according to the protocol described for mammalian and _Drosophila_ S2 cells.

**Whole mount fluorescence staining of _Drosophila_ fat bodies.** _Drosophila_ tissues were stained according to a standard whole mount protocol[46,47]. Briefly, fat bodies freshly dissected in PBS and cultured fat bodies washed in PBS were fixed in 4% paraformaldehyde for 10 min, rinsed three times in PBS for 5 min, permeabilized with 0.1% Triton X-100/PBS for 10 min, and rinsed three times in PBS for 5 min, at 25 °C. Tissues were then incubated with primary antibodies for 14 h at 4 °C, washed three times in PBS for 5 min, incubated with Alexa Fluor secondary antibodies (488, 568, 594, 647, Thermofisher, 1:400 dilution) at 25 °C for 2 h, and with 0.1 μg/ml BODIPY-493/503 at 25 °C for 20 min to stain lipid droplets. After rinsing three times in PBS for 5 min, tissues were incubated with Hoechst 33342 (Life Technologies) and sealed onto glass slides in Prolong mounting medium. Images were acquired with a Nikon ECLIPSE C1si confocal microscope and processed with Nikon NIS-Elements Imaging Software for quantification of lipid droplets (Confocal microscopy facility, University of Trieste). The mean size of lipid droplets was quantified according to Bi and others work[48] in images from $N = 6$ individuals.

**Isolation of mouse mammary epithelial cells.** Mammary glands from 8 to 12-week-old virgin female mice were enzymatically digested and single cell suspensions of purified mammary epithelial cells (MECs) were obtained following a standard protocol[49]. Briefly, mammary glands were digested at 37 °C for 1–2 h in Epi-Cult-B medium (Stem Cell Technologies Inc) with 600 U/ml collagenase (Sigma Aldrich) and 200 U/ml hyaluronidase (Sigma Aldrich). After lysis of the red blood cells with NH$_4$Cl, the remaining cells were washed with PBS/0.02% w/v EDTA. Cells were then dissociated with 0.25% w/v trypsin, 0.2% w/v EDTA for 2 min by gentle pipetting, then incubated in 5 mg/ml Dispase II (Sigma Aldrich) plus 1 μg/ml DNase I (Sigma Aldrich) for 5 min, followed by filtration through a 40 μM cell strainer (BD Falcon). MECs were then purified using the EasySep Mouse Mammary Stem Cell Enrichment Kit (Stem Cell Technologies Inc). MECs were seeded on top of 50 or 0.5 kPa Easy Coat hydrogels (Cell guidance system) coated with 10 μg/ml fibronectin and harvested after 24 h.

**Primary hepatocytes isolation.** Briefly, livers were removed quickly from euthanized mice and the tissue was finely minced with scissors and washed several times with HBSS to remove blood clots. After a last wash with HBSS, the sample was transferred to a falcon with a collagenase solution (0.25 mg/ml Collagenase D in HBSS-Hepes buffer) and was incubated for 45 min at 37 °C in a rotating incubator. At the end of the incubation 10 ml of HBSS containing 5% FBS was added. The cell suspension was filtrated through a 70-μm cell strainer (BD Falcon) and centrifuged at 50 × g for 5 min at 4 °C. The supernatant was discarded, and fresh medium was added. Isolated cells were then seeded on top of 50 kPa or 0.5 kPa Easy Coat hydrogels (Cell guidance system) coated with 10 μg/ml fibronectin and harvested after 24 h.

**Isolation of mouse mesenchymal stem cells.** Mouse mesenchymal stem cells (mMSCs) were isolated from 12-week-old. Mice were anaesthetised, sacrificed and the femurs were harvested. After isolation and cleaning, the marrow was flushed from the bones using 26-gauge needle and 3 mL syringe into complete PBS with 2% FBS and 1 mM EDTA. Cells were gently resuspended and further purified with the EasySep mouse MSC isolation kit (Stem cell 19771), following manufacturer instructions. Viable cells were subsequently resuspended in a growth medium (non-differentiating medium) consisting of DMEM (LONZA), 15% FBS (Euroclone), 2 mM Glutamine, 100 U/mL penicillin and 10 μg/mL streptomycin. Cells were then seeded on cell culture dishes and placed at 37 °C 20% O$_2$. The culture medium was replaced every 2 days until reaching 80% confluence (2 days).

**Differentiation of mouse mesenchymal stem cells in culture.** For spontaneous differentiation on soft matrix, mMSCs were seeded in non-differentiating medium on 50 or 0.5 kPa Easy Coat hydrogels (Cell guidance system) fibronectin-coated hydrogels until reaching 80% confluence (2 days). Then they were cultured in fresh non-differentiating medium or fresh non-differentiating medium supplemented with fatostatin, for 5 days. Cells were then fixed in 4% paraformaldehyde, stained with Oil-Red-O (Sigma Aldrich), rinsed three times with deionised H$_2$O and imaged. For pharmacological induction of adipogenic differentiation on stiff matrix, mMSCs were seeded in non-differentiating medium on plastic culture dishes (Euroclone) until reaching 80% confluence (2 days). Then they were

cultured in either non-differentiating medium, or differentiating medium, consisting of DMEM (LONZA) supplemented with 15% FBS (Euroclone), penicillin 100 U/mL, streptomycin 10 μg/mL, 500 μM 3-isobutyl-1-methylxanthine (Sigma Aldrich), 1 μM dexamethasone (Sigma Aldrich) and 10 μg/ml insulin (Sigma Aldrich), for 3 days. During this period, DMSO, Y-27632 and fatostatin were added daily to differentiating medium. Then, cells were maintained with DMEM, 15% FBS, penicillin 100 U/mL, streptomycin 10 μg/mL, 1 mM glutamine and 10 μg/ml insulin. After seven days, cells were then fixed in 4% paraformaldehyde, stained with Oil-Red-O (Sigma Aldrich) and rinsed three times with deionised $H_2O$ and imaged.

**Animal care**. The mice were housed and used in a specific pathogen-free (SPF) animal facility. Procedures involving animals and their care were performed in conformity with institutional guidelines (D.L. 116/92 and subsequent complementing circulars).

**Analysis of microarray data**. To investigate gene set differentially enriched upon plating on matrixes with different stiffness, we compared the expression profiles of MDA-MB-231 breast cancer cells plated on a stiff substrate (plastic) with the same cells plated on a soft substrate (hydrogels 0.5 kPa). Raw gene expression data were downloaded from Gene Expression Omnibus GSE93529[23]. All data analyses were performed in R (version 3.2.4) using Bioconductor libraries (BioC 3.2) and R statistical packages. Probe level signals were converted to expression values using the robust multi-array average procedure RMA of the Bioconductor affy package. Functional enrichment was performed using Gene Set Enrichment Analysis (http://www.broadinstitute.org/gsea/index.jsp) and gene sets of the Molecular Signature Database (http://software.broadinstitute.org/gsea/msigdb/index.jsp). In particular, we investigated whether the expression levels of MDA-MB-231 cells grown on a soft substrate were associated with elevated expression of the 674 Reactome gene sets. GSEA software was applied on log2 expression data of MDA-MB-231 cells cultured on stiff and soft substrates. Gene sets were considered significantly enriched at FDR < 5% when using Signal2Noise as metric and 1000 permutations of gene sets. The dot plot in Supplementary Fig. 4 was generated using the ggplot2 R package (v.2.2.1, https://cran.r-project.org/web/packages/ggplot2/index.html).

**Analysis of human sample datasets**. To generate a specific SREBP1 transcriptional signature, we selected genes altered in SREBP1 knock-in and SCAP knockout, but not SREBP2-knockin mice[24]. We then compiled a list of the 16 human orthologues of those genes using the DIOPT-DRSC Integrative Ortholog Prediction Tool[50]. We fetched the following datasets from the NCBI GEO database: GSE99621 (from Luzina et al., 2018)[28] and GSE49175 (from Sun et al., 2013)[25]. GSE49175 data were downloaded from the sample table matching each entry, and the log2(mean sample signal/mean signal in reference control) was used as the expression value for each probe. For GSE49175, RNA-seq raw reads were fetched by the NCBI SRA fastq tool[51] and uploaded to the Galaxy web platform, using the public server at usegalaxy.org[52] for analysis. The quality of the reads was assessed by the FastQC tool[53]. The reads were de-interlaced with the FastQ de-interlacer tool[54] and subsequently trimmed using the Trimmomatic[55] with an IlluminaClip step to remove the matching adapter sequences and a sliding-window following step, averaging across 4 bases and requiring average quality of 20. We then used HISAT2[56] to align the reads on the GRChg38 reference human genome in an unstranded manner. Aligned reads were subsequently filtered with SAMtools[57], keeping only aligned reads with minimum MAPQ quality score of 20 and mapped in a proper pair. Aligned reads were counted using featureCounts[58]. Raw data counts were subjected to the Variance Stabilizing Transformation from the DeSeq2 R package (R Development core team, 2008)[59] to obtain normally distributed data for further analyses. The obtained expression data for both datasets were processed by a combined score with zero mean transformation, and the average expression of the SREBP1 signature was calculated averaging the z-score of all the genes in the list. The average value was used as a measure of SREBP1 signature expression. Statistical significance was calculated on the single z-score for all the genes in the signature using a two-way ANOVA test (GraphPad Prism v.7).

**Statistics and reproducibility**. All the experiments are representative of at least three independent repeats. Graph bars represent mean ± s.d. from $n = 3$ biological replicates. All P values were determined using two-tailed Student's t-test, with a 95% confidence threshold as indicated in figure legends.

## Data availability

All data supporting the findings of this study are available from the corresponding author on reasonable request.

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

## Acknowledgements

We thank A. Testa for discussions and proofreading the manuscript. We acknowledge G. Pastore for technical support. We acknowledge support by the Italian Health Ministry (RF-2011-02346976 to G.D.S.), the Italian University and Research Ministry (PRIN-2015-8KZKE3), the Cariplo Foundation (grant no. 2014-0812), Beneficentia-Stiftung, the European Regional Development Fund Interreg Italia-Österreich (PreCanMed ITAT1009), Fondazione CRTrieste and Regione Autonoma Friuli Venezia Giulia (Contributo ex art. 15 L.R. 17/2014) to G.D.S. This work was supported by grants from the Associazione Italiana per la Ricerca sul Cancro (AIRC) and AIRC Special Program Molecular Clinical Oncology '5 per mille' (grant no. 10016) to G.D.S. and S.B., and AIRC IG (grant no. 17659) to G.D.S. M.M. was supported by the FCT Investigator Programme IF/00694/2013 from the Portuguese Foundation for Science and Technology (FCT), Portugal. G.S. is a recipient of a FEBS long-term fellowship. R.B. and M.F. are supported by a FIRC-AIRC fellowship for Italy. F.N. was supported by a European Union FP7/Associazione Italiana per la Ricerca sul Cancro (AIRC) Reintegration Grant (iCARE N° 17885) and a University of Trieste FRA 2018 Starting Grant. We thank the Confocal microscopy facility of the University of Trieste. We also thank F. Feiguin (International Centre for Genetic Engineering and Biotechnology, Trieste, Italy) for providing *Drosophila* cells and lines, and the Vienna Drosophila Resource Center, the Bloomington Drosophila Stock Center, P. Bellosta (University of Trento, Italy), and B. Mollereau (Ecole Normale Superierure de Lyon, France) for providing *Drosophila* lines.

## Author contributions

G.S. and G.D.S. conceived the study. R.B., A.Z., M.F. and G.S. performed the experiments. F.N. and R.B. performed *Drosophila* experiments. M.M. performed the high-content screening. M.F. and S.B. performed bioinformatic analysis. S.M.-S. performed bioinformatics prediction of GGTase1 targets. G.S., R.B., F.N. and G.D.S. designed experiments. G.S., R.B., F.N. and G.D.S. wrote the manuscript.

## Additional information

**Competing interests:** The authors declare no competing interests.

