## [Peer Review File · Nature Communications]

Reviewers' comments:

Reviewer #1 (Remarks to the Author):

This study demonstrates that geranylgeranyl pyrophosphate (GGPP) prevents SREBP1 maturation and function, but not that of SREBP2. Furthermore, ECM stiffness regulates SREBP1 activation via AMPK and RhoA. The experiments in this paper are almost certainly worthy to address the mechanism by which acto-myosin contractility regulates SREBP1 dependent lipid biosynthesis and accumulation, but there is lack of translational application in this study which may limit the general interest to the readership of Nature Communications.

1. The lack of animal disease models harboring genetically manipulated (transgenic or knock out) key molecules, such as SREBP1.
2. Many of the experiments are utilizing LDLR-Luc. However, this more of an SREBP2 canonical downstream target. It is suggested to add canonical SREBP1 downstream targets such as ACC and FAS for these primary experiments.
3. Many experiments use human breast epithelial cells, would liver cells be more applicable?
4. The lack of the human samples validation for the translational application.
5. It is necessary to have statistics for every western blot and immunostain.
6. In Fig. 2a, the authors showed RhoA siRNA is among the top to activate LDLR. However, the top one gene is not addressed. It is suggested to explain the reasoning behind selecting RhoA and not to exploring the function of the first best hit.
7. In fig. 4a and 4c, there is single band of p-AMPK, but there are two bands of p-AMPK in Fig. 4b and 6d. Please explain.
8. In Fig. 4d and 4c, SREBP1 precursor shows different patterns for the bands, one being even throughout treatments and one showing increase with less kPa. Please address the discrepancy.

Reviewer #2 (Remarks to the Author):

This is an interesting and compelling study that provides novel insights into the regulation of SREBP1 (and therefore lipid metabolism and adipogenesis) by the acto-myosin cytoskeleton in response to mechanical cues. The manuscript is clearly written, the data are clearly presented and the interpretation of the data is logical. Overall the findings are of broad interest because they impact the fields of cell signalling (Rho GTPases), lipid metabolism and stem cell differentiation.

Several minor suggestions:

1. p3 1st paragraph: The mevalonate pathway generates a steady-state flux of sterol and isoprenoid biosynthesis. There is little evidence that "increasing the levels of isoprenoids" (ie increasing the flux above normal) is actually a means of controlling biological processes dependent on protein prenylation (ie most prenylated proteins are modified rapidly and constitutively and they are not regulated via the rate of prenylation). However, decreased production of isoprenoids (by a block in the pathway) is clearly deleterious. Some minor rephrasing of this sentence is needed.
2. In places, the text is too vague and needs to be more specific eg p3 last line ("several proteins" actually refers to several hundred that are prenylated!); p8 second para ("several Rho-GTPases).

3. p4 3rd from bottom; what is a "best" hit? The siRNA that had the greatest effect? In Fig. 2A, RhoA scores as the "second best hit" but what was the highest, or even the top 5?
4. p6, 11th line from bottom: it is not clear what effects "they" refers to and this sentence could be made clearer.
5. How do the authors' findings relate to the observations of Liu et al (*Arterioscler Thromb Vasc Biol.* 2002 Jan; 22(1): 76-81), who found that shear stress activated SREBP1 in vascular endothelial cells?

Reviewer #3 (Remarks to the Author):

This manuscript showed that geranylgeranyl pyrophosphate (GGPP) specifically inhibits proteolytic activation of SREBP-1 but not that of SREBP-2. The authors then went on to demonstrate that inhibition of RhoA, a geranylgeranylated protein, activates cleavage of SREBP-1 using various approaches in several in vitro and in vivo systems. Overall, the study is interesting and most data are clear. The major problem of the manuscript is that the authors over interpret their data, and some important conclusions of the study are not supported by the data presented.

1. While the data from figs 2-5 clearly demonstrated that inhibition of RhoA activates cleavage of SREBP-1, these observations by themselves are not sufficient to draw the conclusion that GGPP inhibits SREBP-1 cleavage by stimulating RhoA prenylation to explain the observations shown in Fig. 1. This is because the prenylation status of RhoA was never examined. It is crucial to show that increased geranylgeranylation of RhoA does occur under the same conditions in which GGPP inhibits SREBP-1 cleavage.

2. Mammalian cells can acquire cholesterol through uptake from serum or de novo synthesis. In Fig. 1 the effect of GGPP on SREBP-1 cleavage was determined in cells cultured with statin to deplete endogenously synthesized cholesterol but with the presence of cholesterol in medium, or in medium depleted of cholesterol but still allowed cholesterol synthesis to occur. There was not a single experiment performed under the conditions in which cholesterol was completely depleted. Thus, the data are not sufficient to support the conclusion that the effect of GGPP is sterol-independent. The authors need to examine the effect of sterols on the effect of GGPP more carefully to determine whether this effect is indeed sterol-independent. Alternatively, GGPP may function similarly to PUFA by enhancing the sensitivity of sterols to inhibit SREBP-1 cleavage.

Reviewers' comments:

Reviewer #1 (Remarks to the Author):

This study demonstrates that geranylgeranyl pyrophosphate (GGPP) prevents SREBP1 maturation and function, but not that of SREBP2. Furthermore, ECM stiffness regulates SREBP1 activation via AMPK and RhoA. The experiments in this paper are almost certainly worthy to address the mechanism by which actomyosin contractility regulates SREBP1 dependent lipid biosynthesis and accumulation, but there is lack of translational application in this study which may limit the general interest to the readership of Nature Communications.

1. The lack of animal disease models harboring genetically manipulated (transgenic or knock out) key molecules, such as SREBP1.

We thank the reviewer for the positive comments on our manuscript and we do agree about the impact of using animal models to validate the translatability of our results. However, animal disease models to address the pathological role of SREBP1 are mainly based on transgenic overexpression of mature SREBP1 fragment, which is obviously insensitive to the AMPK inhibitory control (Li et al., 2011, Cell Metabolism 13, 376–388) and thus not very useful for our purposes.

Although we are aware that our manuscript lacks disease models, we think this is not the scope of our work, which, instead, identifies a novel link between mechanical cues and lipid metabolism and is mainly focused on the mechanism of this crosstalk. Nevertheless, by interrogating public datasets of human transcriptional profiles from different physiological and pathological conditions we found, in line with our observations, an inverse correlation between conditions associated with tissue stiffness and SREBP1 transcriptional signature (as a proxy of SREBP1 activity) (see for point 4).

2. Many of the experiments are utilizing LDLR-Luc. However, this more of an SREBP2 canonical downstream target. It is suggested to add canonical SREBP1 downstream targets such as ACC and FAS for these primary experiments.

To address this point, we performed new experiments by using the SREBP1-specific SCD1-Luc construct (Supplementary Figure 1a, 1i, 2d, 2i, 2k). Moreover, as suggested by the Reviewer, we monitored ACC1 and FAS levels in the key experiments of the paper (new Figures 1b, 1h, and Supplementary Figure 1e, 2l).

3. Many experiments use human breast epithelial cells, would liver cells be more applicable?

We have already shown results of experiments performed in mouse primary hepatocytes and in a human liver cell line (Mahlavu) in the first version of the manuscript. However, following the reviewer's advice, we have now introduced a new set of experiments on another human liver cell line, the IHH (Immortalized Human Hepatocytes), and obtained similar results (now shown in new Supplementary figure 1e, f, h, j; Supplementary figure 2e, h, j; Supplementary figure 4a and b).

4. The lack of the human samples validation for the translational application.

To address the reviewer's comment, we generated a transcriptional signature specific for SREBP1 derived from Horton et al., 2003 including genes regulated by SREBP1 and SCAP but not SREBP2, to interrogate two public datasets of human transcriptional profiles from different physiological and pathological conditions: i) normal breast tissue with high vs low mammographic density (Sun et al., 2013 Clin. Cancer Res. 19, 4972–4982), a parameter that directly correlates with tissue stiffness (Boyd et al., 2014 Breast Cancer Res. 16, 417), ii) lung tissue from patients affected by Idiopathic Pulmonary Fibrosis, a condition linked to pulmonary tissue stiffness (Liu et al., 2015 Am. J. Physiol. Cell. Mol. Physiol. 308, L344–L357), vs healthy control (Luzina et al., 2018, Cell Immunology 325, 1–13). Of note, activation of SREBP1 inversely correlated with mammographic density and lung tissue fibrosis, suggesting that, in these physiopathological conditions, SREBP1 might respond to mechanical cues. These data are now included as Figure 3g and in the results section.

5. It is necessary to have statistics for every western blot and immunostain.

We now show in Supplementary Figure 5 the quantification for all the western blots.

6. In Fig. 2a, the authors showed RhoA siRNA is among the top to activate LDLR. However, the top one gene is not addressed. It is suggested to explain the reasoning behind selecting RhoA and not to exploring the function of the first best hit.

We thank the reviewer for his/her comment. As shown in Supplementary table 3, RhoA was the second hit of the screening, while RhoE (also known as ARHE) was the first. However, after validation experiments of the first 5 hits, using siRNA sequences independent from the one used in the screening and two different luciferase reporters (LDLR-Luc and SCD1-Luc), it turned out that RhoA was the actual top hit. We have included these results in Supplementary Fig. 2b-d.

7. In fig. 4a and 4c, there is single band of p-AMPK, but there are two bands of p-AMPK in Fig. 4b and 6d. Please explain.

For western blot analysis, we used phospho-specific AMPK antibody for Thr172 (Cell Signalling 2531S) are known to result in two bands (<https://media.cellsignal.com/pdf/2531.pdf>; Liu et al., 2012, Nature volume 483, pages 608–612). The reason why in some western blots we see a single band could be due to differences in the time of exposure of the film or to the use of gels with different resolution capacity.

8. In Fig. 4d and 4c, SREBP1 precursor shows different patterns for the bands, one being even throughout treatments and one showing increase with less kPa. Please address the discrepancy.

We apologize, the reviewer is right, and the discrepancy was certainly due to over-exposure of the western blot in Figure 4d. To address this point, we run again the western blot, which now shows a pattern of bands consistent with all the others throughout the paper (please see new Figure 4d).

Reviewer #2 (Remarks to the Author):

This is an interesting and compelling study that provides novel insights into the regulation of SREBP1 (and therefore lipid metabolism and adipogenesis) by the acto-myosin cytoskeleton in response to mechanical cues. The manuscript is clearly written, the data are clearly presented, and the interpretation of the data is logical. Overall the findings are of broad interest because they impact the fields of cell signalling (Rho GTPases), lipid metabolism and stem cell differentiation.

We thank this Reviewer for finding our work interesting and compelling and for the positive evaluation of the manuscript.

Several minor suggestions:

1. p3 1st paragraph: The mevalonate pathway generates a steady-state flux of sterol and isoprenoid biosynthesis. There is little evidence that "increasing the levels of isoprenoids" (ie increasing the flux above normal) is actually a means of controlling biological processes dependent on protein prenylation (ie most prenylated proteins are modified rapidly and constitutively and they are not regulated via the rate of prenylation). However, decreased production of isoprenoids (by a block in the pathway) is clearly deleterious. Some minor rephrasing of this sentence is needed.

We agree with the reviewer's comment. To address this point, we changed the sentence as follows: **"Inhibition of SREBPs, by reducing the levels of isoprenoids, impacts on a plethora of biological processes, including cell division, migration, death, intracellular trafficking, protein stability and cytoskeleton organization"**.

2. In places, the text is too vague and needs to be more specific eg p3 last line ("several proteins" actually refers to several hundred that are prenylated!); p8 second para ("several Rho-GTPases).

The complete sentence in the p.3 of the original version of the manuscript is "the primary role of GGPP is to control subcellular localization and function of several proteins". Based on our prediction that there are 75 proteins predicted to be geranylgeranylated, we modified the text and changed the term "several" to "considerable number".

3. p4 3rd from bottom; what is a "best" hit? The siRNA that had the greatest effect? In Fig. 2A, RhoA scores as the "second best hit" but what was the highest, or even the top 5?

The results of the screening are shown in Supplementary Table 3. RhoA was the second hit of the screening, while RhoE (also known as ARHE) was the first. However, after validation experiments of the first 5 hits, using siRNA sequences independent from the one used in the screening and two different luciferase reporters (LDLR-Luc and SCD1-Luc), it turned out that RhoA was the actual top hit. We have included these results in Supplementary Fig. 2b-d.

4. p6, 11th line from bottom: it is not clear what effects "they" refers to and this sentence could be made clearer.

We agree with the reviewer; therefore, we changed the text as follows: "We tested this hypothesis by growing human epithelial breast cells on fibronectin-coated hydrogels characterized by progressively reduced elastic moduli (e.g. 50, 4 and 0.5 kPa; Fig. 3a). In these conditions, ECM softening led to a marked impairment of mechano-signaling pathways, as demonstrated by reduction of phosphorylated MLC2 (pMLC2) and Focal Adhesion Kinase (pFAK), as well as of the protein levels of the mechano-transducer TAZ (Fig. 3b, e), and triggered a progressive induction of SREBP1 protein maturation (Fig. 3b, e) and transcriptional activity (Fig. 3c, d and Supplementary Fig 4a, b). Furthermore, cells grown in soft ECM showed a marked SREBP1-dependent lipid droplets accumulation (Fig. 3d, f and Supplementary Fig 4b).".

5. How do the authors' findings relate to the observations of Liu et al (*Arterioscler Thromb Vasc Biol.* 2002 Jan;22(1):76-81), who found that shear stress activated SREBP1 in vascular endothelial cells?

Liu et al. investigated the effect of shear stress on SREBP1 activation in vascular endothelial cells. They showed a positive effect of the shear stress on SREBP1 nuclear localization and activation. However, this effect was fast and transient, being completely lost already 4 hours after the application of the shear stress (please see Figure 1a in Liu et al paper). Further, 12 hours after the stimulus, SREBP1 was completely localized in the cytoplasm (please see Figure 3 in Liu et al paper), thus suggesting that long-term application of this type of mechanical stress inhibits SREBP1. Despite the differences in the biological context and in the mechanical stimulus applied to the cells, we believe that the data in Liu et al. are in line with our experimental evidence showing that growing cells on a soft ECM for 16h triggers SREBP1 activation.

Reviewer #3 (Remarks to the Author):

This manuscript showed that geranylgeranyl pyrophosphate (GGPP) specifically inhibits proteolytic activation of SREBP-1 but not that of SREBP-2. The authors then went on to demonstrate that inhibition of RhoA, a geranylgeranylated protein, activates cleavage of SREBP-1 using various approaches in several in vitro and in vivo systems. Overall, the study is interesting, and most data are clear. The major problem of the manuscript is that the authors over interpret their data, and some important conclusions of the study are not supported by the data presented.

We thank this Reviewer for his/her positive evaluation of our manuscript and we kept in great consideration the comments when interpreting our results in the new version of the MS.

1. While the data from figs 2-5 clearly demonstrated that inhibition of RhoA activates cleavage of SREBP-1, these observations by themselves are not sufficient to draw the conclusion that GGPP inhibits SREBP-1

cleavage by stimulating RhoA prenylation to explain the observations shown in Fig. 1. This is because the prenylation status of RhoA was never examined. It is crucial to show that increased geranylgeranylation of RhoA does occur under the same conditions in which GGPP inhibits SREBP-1 cleavage.

We agree with the Reviewer's comment. In Figure 1a we show that adding back GGPP to statin-treated breast epithelial cells (MCF 10A) cells prevented SREBP activation and we suggest that this effect was dependent on RhoA prenylation. To demonstrate that GGPP addition re-established RhoA prenylation, we evaluated the RhoA-prenylation upon statin treatment. As shown in new Figure 2c, statin treatment reduced RhoA prenylation (assessed by immunoprecipitating endogenous RhoA followed by immunoblot using an antibody to reveal the farnesyl and geranylgeranyl moieties). The results of this experiments showed that GGPP addition to cerivastatin-treated cells rescued RhoA prenylation and prevented SREBP1 activation. The observed changes in the levels of prenylated RhoA corresponded to geranylgeranylated form as demonstrated by the effect that GGTI-298 but not FTI treatment reduced RhoA prenylation and prevented SREBP1 activation. The text has been modified accordingly.

2. Mammalian cells can acquire cholesterol through uptake from serum or de novo synthesis. In Fig.1 the effect of GGPP on SREBP-1 cleavage was determined in cells cultured with statin to deplete endogenously synthesized cholesterol but with the presence of cholesterol in medium, or in medium depleted of cholesterol but still allowed cholesterol synthesis to occur. There was not a single experiment performed under the conditions in which cholesterol was completely depleted. Thus, the data are not sufficient to support the conclusion that the effect of GGPP is sterol-independent. The authors need to examine the effect of sterols on the effect of GGPP more carefully to determine whether this effect is indeed sterol-independent. Alternatively, GGPP may function similarly to PUFA by enhancing the sensitivity of sterols to inhibit SREBP-1 cleavage.

We agree with the Reviewer, this is a key point indeed. Following his/her advise, we performed new experiments by maintaining statin-treated cells in lipid-depleted medium and, as such, completely depleted of cholesterol. Of note, in this experimental set-up, GGPP addition to the culture medium prevented SREBP1 maturation and activity, thus undoubtedly proving that GGPP regulate SREBP1 in a cholesterol-independent manner (Figure 1d and Supplementary figure 1b-d). We have modified the text accordingly.

REVIEWERS' COMMENTS:

Reviewer #1 (Remarks to the Author):

This group of authors have addressed most, if not all of my critics. New experiments have been performed following reviewers' suggestions. The newly acquired data have been incorporated to the existing results tightly. I have no further comments.

Reviewer #2 (Remarks to the Author):

The authors have robustly addressed all of the comments of each reviewer in detail, by providing a considerable amount of new data that supports the initial studies and clarification of some areas of the text. These new data are included in the main figures and also as supplementary figures. Overall these strengthen the manuscript and provide an even more compelling argument for a role of mechanical/cytoskeletal forces in regulating SREBP1 and lipid metabolism.

Reviewer #3 (Remarks to the Author):

The revised manuscript addresses my previous concerns.